# InstructNav: Zero-shot System for Generic Instruction Navigation in Unexplored Environment

**Yuxing Long**[123*], **Wenzhe Cai**[4*], **Hongcheng Wang**[123], **Guanqi Zhan**[5] **and Hao Dong**[123†]

[1]CFCS, School of Computer Science, Peking University   [2]PKU-Agibot Lab
[3]National Key Laboratory for Multimedia Information Processing, School of Computer Science,
Peking University [4]School of Automation, Southeast University   [5]University of Oxford

**Abstract:** Enabling robots to navigate following diverse language instructions in unexplored environments is an attractive goal for human-robot interaction. However, this goal is challenging because different navigation tasks require different strategies. The scarcity of instruction navigation data hinders training an instruction navigation model with varied strategies. Therefore, previous methods are all constrained to one specific type of navigation instruction. In this work, we propose InstructNav, a generic instruction navigation system. InstructNav makes the first endeavor to handle various instruction navigation tasks without any navigation training or pre-built maps. To reach this goal, we introduce Dynamic Chain-of-Navigation (DCoN) to unify the planning process for different types of navigation instructions. Furthermore, we propose Multi-sourced Value Maps to model key elements in instruction navigation so that linguistic DCoN planning can be converted into robot actionable trajectories. With InstructNav, we complete the R2R-CE task in a zero-shot way for the first time and outperform many task-training methods. Besides, InstructNav also surpasses the previous SOTA method by 10.48% on the zero-shot Habitat ObjNav and by 86.34% on demand-driven navigation DDN. Real robot experiments on diverse indoor scenes further demonstrate our method's robustness in coping with the environment and instruction variations. The project webpage is https://sites.google.com/view/instructnav.

**Keywords:** Generic Instruction Navigation, Zero-shot, Unexplored Environment

## 1  Introduction

Instructing the robot to navigate in the unexplored indoor scene via natural language is user-friendly, which has drawn considerable interest within the robotic community [1][2][3][4][5]. A plausible future direction is to develop the robot that can understand and execute a large variety of instructions. This will greatly expand the application scenarios of instruction navigation robots. However, different types of instructions emphasize different navigation strategies. For example, object goal navigation approaches [6][7] primarily concentrate on performing efficient exploration to find the target object in unseen environments; visual language navigation methods [8][9] focus on following step-by-step instruction; demand-driven navigation [5] works are geared towards conducting demand-based commonsense reasoning. There are significant differences among their navigation strategies, which makes training an instruction navigation model that can follow diverse instructions difficult. The scarcity of instruction navigation data exacerbates this difficulty. Therefore, almost all previous works are limited to executing one type of navigation instructions and cannot adapt to other types. This raises the question - ***Can we develop a zero-shot system for generic instruction navigation in the unexplored environment?***

The first challenge lies in how to unify different types of instructions. To reach this goal, we propose a generic navigation planning paradigm called **Dynamic Chain-of-Navigation (DCoN)**. The

---

* Joint first authors † Corresponding Author (hao.dong@pku.edu.cn)

8th Conference on Robot Learning (CoRL 2024), Munich, Germany.

DCoN models critical elements in navigation - actions and landmarks, as well as their consequential relationship. It inherently corresponds to the Chain-of-Thought thinking process of the large language model (LLM). This alignment allows the LLM to convert navigation instructions to DCoN without manual annotations. More importantly, DCoN is not a static and simple instruction decomposition but a generic navigation strategy that updates with the newly explored environment. During the navigation, the next navigation action and landmarks in DCoN will be dynamically updated at every decision-making step with observed objects considering LLM's inner commonsense about room layout and human habits. This way, DCoN can inspire InstructNav to align semantic labels, efficiently explore the unseen environment, and conduct commonsense reasoning about landmarks.

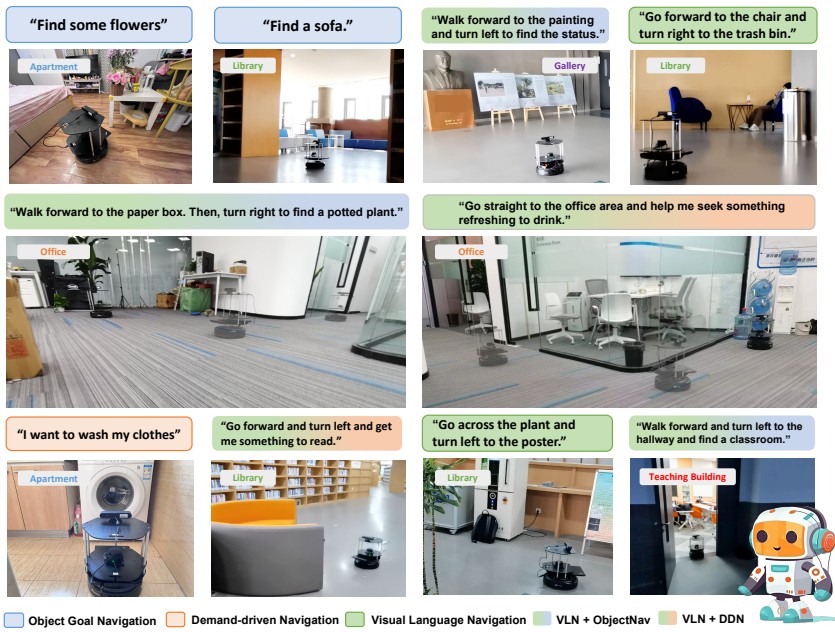

Figure 1: InstructNav can follow different types of navigation instructions in diverse indoor scenes.

With unified DCoN planning, the ensuing challenge is how to control the robot by this linguistic planning. To address this problem, we propose **Multi-sourced Value Maps**, which represents key elements in instruction navigation, including action, landmark, and history trajectory, on four value maps to decide the next waypoint and actionable trajectories. The Action Value Map and Semantic Value Map are created according to the next DCoN action and landmarks. They can encourage the robot to follow the specified action and move toward navigation landmarks. The Trajectory Value Map is established based on the navigation trajectory to avoid repetitive movement. Although these value maps can deal with simple instructions, there still exist difficulties when navigation instructions require multimodal reasoning. For example, the Semantic Value Map struggles to figure out which table on the map is "*the front dining table*" while the Action Value Map fails to represent "*walk between*". To improve InstructNav's capability on multimodal reasoning, we further design an Intuition Value Map. The multimodal large model [10]'s prediction on the next navigation area is projected on the Intuition Value Map to guide the robot. At each decision step, these value maps are synthesized to plan the next waypoint. Their collaboration form InstrcutNav's generic instruction navigation capability, which single semantic map-based methods like [11] [12] cannot realize.

In this work, our primary contribution is InstructNav, the first generic instruction navigation system that can execute different types of instructions in the continuous environment without any navigation training or pre-built maps. To reach this goal, we introduce Dynamic Chain-of-Navigation to unify different types of navigation instructions into a standard planning paradigm. We create Multi-sourced Value Maps to model the effect of key elements in instruction navigation. This way, the linguistic DCoN planning can be converted into robot actionable trajectories. With InstructNav, we first achieve zero-shot performance on the R2R-CE task [3] and outperform a wide array of task-training methods. Furthermore, InstructNav achieves 10.48% and 86.34% improvement on the zero-shot

Habitat ObjNav and demand-driven navigation DDN compared with state-of-the-art models. In the real world, InstrtuctNav demonstrates robustness in diverse kinds of instructions and indoor scenes including apartment, office, library, gallery, and teaching buildings.

## 2 Related Work

### 2.1 Instruction-guided Navigation

Controlling a navigation robot through natural language instructions in unexplored environments is a user-friendly interaction mode. Many research works have been conducted in this area. According to the instruction type, these works can be categorized into object goal navigation, visual language navigation, and demand-driven navigation. The object goal navigation methods [13][1][6][7][14][15][16][17][18] can find one specific object in the scene. The visual language navigation approaches [3][15][16][17][8][9] can follow one step-by-step instruction to reach a specified destination. The demand-driven navigation models [5] can satisfy human demand by searching for related objects in the scene. Although these methods can perform a pre-defined type of instruction, all of them are incapable of executing different kinds of instructions, which significantly limits their application scenarios. Compared with them, our InstructNav can follow diverse types of instructions to navigate with convincible success rates.

### 2.2 Large Models in Robotics Navigation

With internet-scale training data, large models including large language models [19][20][21][22] and multimodal large models [23][24][25] [26][27] have emerged with powerful capabilities, which include instruction following, task planning, and visual perception. These capabilities are closely related to instruction navigation. The development of large models has motivated their application in robotic navigation. Some [12][6][28][29][11] directly used visual perception features encoded by multimodal large models (*e.g.*, CLIP[23] and BLIP2[24]) to retrieve target position while others [30][31][32][33][18] just simply leveraged large language models to make high-level navigation planning. Although these methods are all based on large models with strong generalization capabilities, none of them support different kinds of navigation instructions. Our InstructNav aims to relive the potential of large models to navigate with different kinds of instructions in a zero-shot way.

## 3 METHODOLOGY

### 3.1 Problem Formulation and Method Overview

**Problem Formulation** The generic instruction navigation task requires the robot to follow one instruction $I$ in natural language format to reach the target location in the unexplored continuous environment. At each step, the robot can observe egocentric RGB image $V_t$ and depth image $D_t$. The robot also knows its camera pose $P_t$. With observations $O_t = \{V_t, D_t, P_t\}$, the robot needs to execute low-level action $a_t$ to move towards the target. No pre-built maps are allowed in this task.

**Method Overview** We propose to handle the unified instruction navigation via a new planning paradigm: Dynamic Chain-of-Navigation (DCoN) in Section 3.2. Then, we create Multi-sourced Value Maps (Section 3.3) to model key elements in generic instruction navigation. By deciding with these value maps, our method can plan actionable trajectories for low-level movement (Section 3.4).

### 3.2 Dynamic Chain-of-Navigation Planning

After analyzing instruction-guided navigation processes, we discern that the navigation robot should follow one action to move towards specific navigation landmarks. Consequently, the navigation instructions can be transformed into an "*Action 1 - Landmark 1 → Action 2 - Landmark 2 → ...*" schema, which is similar to the chain-of-thought thinking process of the large language model [34]. Therefore, this paradigm can be aptly named "Chain-of-Navigation (CoN)". The conversation from raw navigation instruction to CoN can be obtained through LLM.

However, generic navigation planning cannot be realized by directly extracting actions and landmarks from raw instructions. Firstly, the landmarks specified in the instruction like "*arched wooden doors*" may not align with the object labels produced by the semantic segmentation model like "*Doorway*",

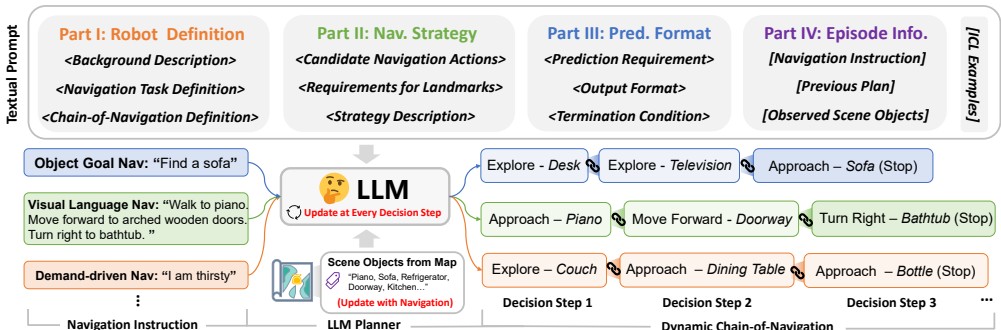

Figure 2: **The workflow of Dynamic Chain-of-Navigation (DCoN).** Different types of navigation instructions can be unified into DCoN by LLM. The next action and landmarks will be updated based on observed scene objects at every decision step. Beyond extracting actions and landmarks, DCoN achieves semantic label alignment, common-sense reasoning, and environmental exploration for navigation planning. Complete prompt and output examples are shown in *Supplementary Materials*.

which hinders the retrieval of landmarks on the map. Secondly, the target landmarks may not be observed in the explored environment. At this time, navigation planning should inspire efficient environment exploration. Thirdly, instruction with abstract human demand like "*I am thirsty*" cannot be decomposed into concrete actions and landmarks at all.

To address these problems, we propose Dynamic Chain-of-Navigation (DCoN) as shown in Figure 2. Rather than make the whole plan at one time, the DCoN will be re-inferred at each decision-making step to update the next action and landmarks based on observed scene objects. The textual prompt for DCoN is composed of "Robot Definition", "Navigation Strategy", "Prediction Format" and "Episode Information" four parts. The <*Candidate Navigation Actions*> defines common navigation actions like "*Explore*", "*Approach*", "*Move Forward*" ... to choose from. The <*Requirements for Landmarks*> prioritizes the observed objects when planning the next landmarks so that they are more likely to be retrieved in the creation of the Semantic Value Map. Strategies for different instruction navigation tasks are defined in <*Strategy Description*> to infer the next landmark considering given navigation instruction, common house layout, and human habits. The LLM's prediction is formatted as {*'Reason':... 'Action':... 'Landmark':... 'Flag':...*}. As Figure 2 examples, DCoN can align "*arched wooden doors*" with "*Doorway*", guide the robot to explore areas with *TV* to find a *sofa* and approach *a bottle of water* to relieve *thirst*. With DCoN, InstructNav realizes landmark alignment, environment exploration, and commonsense reasoning for different types of navigation instructions.

### 3.3 Multi-sourced Value Maps

The DCoN planning is in language format, which cannot directly control the robot's movements. To convert the linguistic DCoN planning into robot actionable trajectories, we create Multi-sourced Value Maps (Figure 3). These value maps model key factors in the generic instruction navigation, including action, landmarks, and navigation history. The next waypoint and actionable trajectories can be decided based on Multi-sourcecd Value Maps. During the navigation, RGB-D and camera pose are utilized to build scene point cloud $PCD$. The area on the explored ground, free from obstacles, is selected as navigable area $PCD_{nav}$. The next DCoN action and landmarks to be executed are denoted as $A_i$ and $L_i$. Each value map is initialized with all-zero values and detailed in the following.

#### 3.3.1 Semantic Value Map

Accurately mapping observed objects' locations and semantic information is critical for navigating toward DCoN-specified landmarks. To this end, we create a Semantic Value Map $m_s$ for DCoN landmarks $L_i$. In the navigation process, the 2D semantic segmentation mask [35] is lifted to 3D by depth and camera pose to obtain scene semantic point cloud $PCD_{obj}$. The semantic value $C_{sem}$ of $m_s$ is calculated based on the normalized minimal distances between each navigable area position $p \in PCD_{nav}$ and the $L_i$ positions $q \in PCD_{obj}$. following Equation 1 and 2.

$$d_{sem} = \min_{q \in PCD_{obj}} \|p - q\|, \ \forall p \in PCD_{nav} \tag{1}$$

$$C_{sem} = 1 - \frac{d_{sem} - \min(d_{sem})}{\max(d_{sem}) - \min(d_{sem})} \tag{2}$$

As the above equations, the areas near the landmarks $L_i$ will have relatively higher values than others.

### 3.3.2 Action Value Map

To endow InstructNav with the capability to execute concrete movement actions and explore the frontiers, we propose the Action Value Map $m_a$. A value assignment operation is conducted on the initial all-zero value map by the DCoN $A_i$ type. The operation specifics are detailed below:

- *Move forward / Turn around / Turn right / Turn left*: Values of one are assigned to the front / back / right / left sector area at the robot current location. Each region corresponds to one-quarter of the panoramic Field of View (FOV) at the current location.

- *Explore*: Values of one are set for the boundaries of the present explored environment.

- *Enter / Exit*: Given that entering or exiting any room necessitates crossing door-shaped regions, the "*Enter*" or "*Exit*" actions will be replaced by a "*Approach*" action and a "*Doorway*" landmark will be integrated into the DCoN planning.

- *Approach*: This action does not necessitate any operation on the Action Value Map. It can be realized directly through the Semantic Value Map.

This way, the action $A_i$ related areas will have higher values than others on $m_a$.

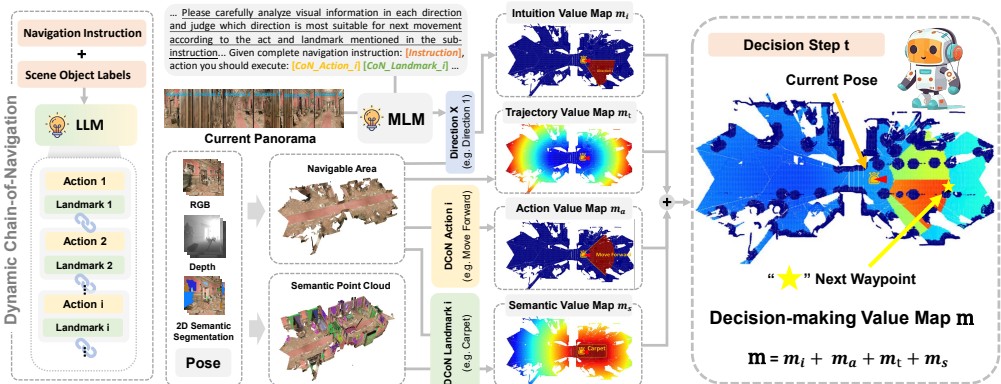

Figure 3: **The system framework of InstructNav.** The next *Action i* and *Landmarks i* are obtained from DCoN. Scene semantic point cloud is created from the RGB-D observation and 2D semantic segmentation. With this information, Multi-sourced Value Maps $m_a, m_s, m_t$, and $m_i$ can be established. Areas with **redder** colors represent higher ↑ values, while **bluer** colors indicate lower ↓ values. By synthesizing them into a decision-making value map $m$, InstructNav can plan the next waypoint.

### 3.3.3 Trajectory Value Map

To encourage more diverse navigation trajectories, we design a Trajectory Value Map $m_t$ for Instruct-Nav. During the navigation, the robot's positions are continually recorded as a history trajectory $PCD_{traj}$. The trajectory value $C_{traj}$ of $m_t$ is calculated as the normalized minimal distances between each navigable area position $p \in PCD_{nav}$ and every history position $h \in PCD_{traj}$.

$$d_{traj} = \min_{q \in PCD_{traj}} \|p - h\|, \ \forall p \in PCD_{nav} \tag{3}$$

$$C_{traj} = \frac{d_{traj} - \min(d_{traj})}{\max(d_{traj}) - \min(d_{traj})} \tag{4}$$

As Equation 3 and 4, the areas far from history trajectory have relatively higher values to navigate.

### 3.3.4 Intuition Value Map

To further improve InstructNav's capabilities on multimodal semantic reasoning, we propose an Intuition Value Map $m_i$. The next navigation area predicted by the multimodal large model (MLM) is projected on this map with values of one to guide the robot to move. Figure 3 displays how InstructNav leverages GPT-4V to predict the next navigation area. For the visual input, $N$ RGB

observations from different directions at the current position are sampled at equal intervals and concatenated into a panorama image $P_i$. The effect of $N$ is studied in the Ablation Study. Each observation image is annotated with the corresponding direction ID. For the textual input, we define the task and provide a complete navigation instruction $I$, next action $A_i$, and landmarks $L_i$ from Dynamic Chain-of-Navigation. In response, the large model will first conduct chain-of-thought $CoT_i$ to analyze visual information in each direction and then make decisions on the next movement direction $Dir_i$. Refer to *Supplementary Materials* for details about input prompt and output examples. The multimodal large model makes inferences as Equation 5.

$$(CoT_i, Dir_i) = MLM(P_i; I, A_i, L_i) \tag{5}$$

The FOV at direction $Dir_i$ is projected on the Intuition Value Map as MLM predicted navigation area. If there are navigable positions in this area, they will be assigned values of one. Otherwise, the failure feedback will be transmitted back to the large model, instigating re-prediction.

### 3.4 Navigation Process with Multi-sourced Value Maps

As Equation 6, a decision-making value map $m$ is obtained by summing over all four value maps.

$$m = m_i + m_a + m_t + m_s \tag{6}$$

Obstacles areas on the decision-making value map $m$ are set to zero for obstacle avoidance. Then, following Equation 7, the navigation goal $(x_i, y_i, z_i)$ is set to be the point with the highest value on the $m$ and the robot trajectory is planned by A* algorithm, which is also based on the generated value map $m$. In the simulator, as the action space follows a discrete setting, we use a simple rotate-then-forward[36] to track the planned path and while in the real-world, we directly control the robot speed. The navigation is stopped when DCoN sets "*Flag*" to "*True*" or GPT-4V outputs "*Stop*" judgment.

$$(x_i, y_i, z_i) = \arg\max_{(x,y,z) \in m} P(x, y, z) \tag{7}$$

Table 1: **Comparison with SOTA methods on HM3D object goal navigation.**

| Method | Training Free | Novel Object | Support Instruction | Navigation Error | Success Rate | SPL |
|---|---|---|---|---|---|---|
| SemExp [37] | ✗ | ✗ | ObjectNav | 2.94 | 37.9 | 18.8 |
| PixelNav [7] | ✗ | ✓ | ObjectNav | - | 37.9 | 20.5 |
| Habitat-Web [38] | ✗ | ✗ | ObjectNav | - | 41.5 | 16.0 |
| OVRL [39] | ✗ | ✗ | ObjectNav | - | **62.0** | **26.8** |
| ZSON [40] | ✓ | ✓ | ObjectNav | - | 25.5 | 12.6 |
| ESC [6] | ✓ | ✓ | ObjectNav | - | 39.2 | 22.3 |
| VoroNav [41] | ✓ | ✓ | ObjectNav | - | 42.0 | 26.0 |
| L3MVN [14] | ✓ | ✓ | ObjectNav | 4.43 | 50.4 | 23.1 |
| VLFM [11] | ✓ | ✓ | ObjectNav | - | 52.5 | **30.4** |
| InstructNav (Ours) | ✓ | ✓ | **Generic** | **2.58** | **58.0** | 20.9 |

Table 2: **Comparison with SOTA methods on R2R-CE visual language navigation. For fairness, only methods free from MP3D dataset-specific waypoint predictors are studied.**

| Method | Training Free | Support Instruction | Trajectory Length | Navigation Error | Success Rate | SPL |
|---|---|---|---|---|---|---|
| Sasra [42] | ✗ | VLN | 7.89 | 8.32 | 24 | 22 |
| Seq2Seq [3] | ✗ | VLN | 9.30 | 7.77 | 25 | 22 |
| CWP-CMA [8] | ✗ | VLN | 8.22 | 7.54 | 27 | 25 |
| CWP-RecBERT [8] | ✗ | VLN | 7.42 | 7.66 | 23 | 22 |
| Ego2Map-NaViT [9] | ✗ | VLN | 8.03 | 7.25 | 30 | 29 |
| CMA [3] | ✗ | VLN | 8.64 | 7.37 | 32 | 30 |
| NaVid [43] | ✗ | VLN | 7.63 | **5.47** | **37** | **36** |
| InstructNav (Ours) | ✓ | **Generic** | 7.74 | **6.89** | **31** | **24** |

Table 3: **Comparison with SOTA methods on DDN demand-driven navigation.**

| Method | Training Free | Novel Object | Support Instruction | Trajectory Length | Success Rate | SPL |
|---|---|---|---|---|---|---|
| ZSON-demand [5] | ✗ | ✓ | DDN | - | 3.5 | 2.4 |
| VTN-CLIP-demand [5] | ✗ | ✓ | DDN | - | 9.3 | 3.9 |
| DDN [5] | ✗ | ✓ | DDN | 3.85 | 16.1 | 8.4 |
| ChatGPT-Prompt [19] | ✓ | ✓ | DDN | 0.60 | 0.3 | 0.01 |
| MiniGPT-4 [44] | ✓ | ✓ | DDN | 0.74 | 2.9 | 2.0 |
| InstructNav (Ours) | ✓ | ✓ | **Generic** | 4.44 | **30.0** | **14.2** |

# 4 EXPERIMENTS

## 4.1 Experiment Setup

For object goal navigation, we evaluate our method on the HM3D [45] dataset in Habitat simulator following Habitat ObjectNav challenge's setting [1]. For visual language navigation, we test our method on the R2R-CE [3] dataset val-unseen split with the Habitat simulator. For demand-driven navigation, we evaluate our method on the DDN [5] dataset based on AI2Thor [46] simulator and ProcThor [47] scenes following its unseen scenes and instructions setting. As previous work [3] [1] [5], we take Trajectory Length (TL), Navigation Error (NE), Success Rate (SR), Oracle Success Rate (OSR), and SR penalized by Path Length (SPL) as evaluation metrics.

## 4.2 Implementation Details

We utilize GPT-4 (*gpt-4-0613*) [21] to plan Dynamic Chain-of-Navigation and adopt GPT-4V (*gpt-4-1106-vision-preview*) [10] to judge navigation directions for Intuition Value Map. Their parameters are all at the OpenAI default settings. The number of RGB observations in the visual prompt (*i.e.*, $N$) is set to 6 following the ablation study result. When creating semantic point cloud, we deploy GLEE [35] on one RTX 4090 GPU to conduct semantic segmentation on the RGB image. Note that our InstructNav is completely free from navigation training and pre-built maps.

## 4.3 Simulation Experiments

**Object Goal Navigation on HM3D** We compare our method with state-of-the-art object goal navigation models on HM3D datasets. Table 1 shows that our InstructNav outperforms all zero-shot methods on success rate and is comparable to the best trained object navigation model OVRL [39].

**Visual Language Navigation on R2R-CE** Many previous methods rely on waypoint predictor [36] [48] specifically trained on the Matterport3D topological map to improve their performances on MP3D-related navigation tasks. Our method predicts the next waypoint based on our designed Multi-sourced Values Maps in a zero-shot way. Therefore, We follow [8], [9] and [43] to compare our InstructNav with VLN models free from waypoint predictor trained on MP3D topological map. From Table 2, it can be observed that InstructNav is the first model that completes visual language navigation task in a zero-shot way and it outperforms a wide array of task-trained models.

**Demand-driven Navigation on DDN** The baselines of the DDN task include task-adapted large models. We compare our method with these trained and non-trained models on the unseen scenes and unseen instructions. As Table 3, InstructNav outperforms all baselines by a large margin.

### 4.3.1 Ablation Study

To conduct the ablation study, we randomly sampled 100 instructions for each task respectively.

**The effect of $N$ RGB observations in MLM's visual prompt**

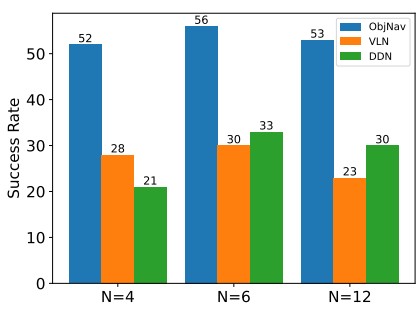

Figure 4: Effect of $N$ RGB observations.

We test three different values of $N$ to study its influence on InstructNav. The "$N = 4$" concatenates front, right, back, and left (*i.e.*, Direction 1, 4, 7, 10) RGB observations as MLM's visual prompt while "$N = 12$" concatenates all twelve RGB observations. The "$N = 6$" selects 6 out of all 12 RGB observations with an interval 1 as visual input to MLM. From Figure 4, we can observe that "$N = 6$" consistently produces better performance on all three navigation tasks. After analyzing failure cases, we found that smaller $N$ may miss some key visual information, while larger $N$ may increase the understanding burden of MLM. Therefore, we set $N$ to 6 in our implementation.

**The effect of DCoN and Multi-sourced Value Maps** To verify the effectiveness of DCoN and Multi-sourced Value Maps in our method, we ablate DCoN and four value maps in Table 4. Through the experiment, we find that DCoN is critical to the InstructNav. Ablating DCoN results in a significant decrease in the success rate on all three tasks. Our method relies on DCoN to achieve unified planning for different instruction navigation tasks. Besides, all four Multi-sourced Value

Maps show improvement as expected. The ablation of any one value map will weaken InstructNav's performance on all tasks. The Multi-sourced Value Map is a suitable representation of key factors involved in the decision of generic instruction navigation.

Table 4: **Ablation study about DCoN and Multi-sourced Value Maps in our InstructNav system.**

| Method | Object Nav. on HM3D | | | Visual Language Nav. on R2R-CE | | | | | Demand-driven Nav. on DDN | | |
|---|---|---|---|---|---|---|---|---|---|---|---|
| | NE | SR | SPL | TL | NE | OSR | SR | SPL | TL | SR | SPL |
| InstructNav | **2.91** | **56** | **22.5** | 6.74 | **6.04** | 42 | **30** | 22 | 4.27 | **33** | 15.6 |
| w/o Dynamic CoN (DCoN) | 3.09 | 44 | 19.4 | 7.42 | 7.49 | 46 | 23 | 18 | 4.51 | 22 | 10.8 |
| w/o Action Value Map | 3.26 | 51 | 19.1 | 6.82 | 6.97 | 39 | 28 | 22 | 4.76 | 28 | **15.9** |
| w/o Semantic Value Map | 3.17 | 44 | 18.9 | 8.13 | 8.64 | 40 | 21 | 17 | 5.01 | 25 | 14.1 |
| w/o Trajectory Value Map | 3.00 | 52 | 21.7 | 5.54 | 6.83 | 31 | 19 | 16 | 3.81 | 21 | 11.4 |
| w/o Intuition Value Map | 3.13 | 54 | 21.2 | 9.22 | 9.18 | **47** | 17 | 11 | 5.62 | 20 | 11.1 |

**The effect of open-source large models** We further replace GPT models in InstructNav with Llama3 70B [22] and LLaVA1.6 34B (*i.e.*, LLaVA-NeXT 34B) [27] to explore the feasibility of driving InstructNav by open-source models. Specifically, Llama3 is utilized to plan the DCoN while LLaVA1.6 is adopted to create the Intuition Value Map. From Table 5, we can observe that open-source models can achieve comparable performances to GPT models in a portion of tasks (*e.g.*, ObjectNav). This benefits from the robust design of DCoN and Multi-sourced Value Maps in the InstructNav. However, open-source models still exhibit weakness compared with GPT models.

Table 5: **Comparison between open-soured and close source large models in InstructNav.**

| Large Models in InstructNav | Object Nav. on HM3D | | | Visual Language Nav. on R2R-CE | | | | | Demand-driven Nav. on DDN | | |
|---|---|---|---|---|---|---|---|---|---|---|---|
| (Language + Multimodal) | NE | SR | SPL | TL | NE | OSR | SR | SPL | TL | SR | SPL |
| GPT4 + GPT4V | **2.91** | **56** | **22.5** | 6.74 | **6.04** | **42** | **30** | **22** | 4.27 | **33** | **15.6** |
| Llama3 70B + GPT4V | 3.19 | 50 | 18.9 | 6.89 | 7.02 | 40 | 23 | 19 | 4.09 | 21 | 11.0 |
| GPT4 + LLaVA1.6 34B | 3.26 | 50 | 19.4 | 6.12 | 7.97 | 26 | 17 | 13 | 5.45 | 28 | 13.9 |
| Llama3 70B + LLaVA1.6 34B | 3.14 | 50 | 17.8 | 5.82 | 8.34 | 24 | 12 | 9 | 5.30 | 18 | 8.3 |

## 4.4 Real Robot Experiments

We perform real robot experiments based on the *Turtlebot 4* mobile robot. Our robot is equipped with an *ORBBEC Astra Pro Plus* RGB-D camera connected to the *ThinkPad E14* laptop computer and an *RPLIDAR-A1* lidar connected to *Raspberry Pi 4B* as sensors. All processors are installed on the robot and communicate with each other through a portable WiFi. To accelerate, InstructNav is deployed on a remote RTX 4090 workstation. We utilize the SLAM Toolbox [49] for self-localization and the Navigation2 [50][51] for point-to-point navigation with dynamic obstacle avoidance.

The real robot experiments are conducted in representative indoor scenes including large-scale offices and multi-room apartments, open library, gallery, and teaching building. To demonstrate the effectiveness of our approach, every instruction is independently executed without any pre-built map. For each scene, we create diverse navigation instructions covering different instruction types and navigation goals. Figure 1 displays the scenes and a part of navigation instructions in our real robot experiments. Videos about real robot experiments can be found in the *Supplementary Materials*.

## 5 Conclusion

In this work, we focus on developing the first generic instruction navigation system InstructNav in the continuous environment without any navigation training or pre-built maps. To reach this goal, we propose the Dynamic Chain-of-Navigation to unify different navigation instructions and model key elements in instruction navigation through Multi-sourced Value Maps. This way the linguistic DCoN planning can be converted into robot actionable trajectories. Extensive experiments on the simulators and the real robot demonstrate the generalization and effectiveness of our training-free method.

**Limitation and Future work** The current InstructNav system still relies on close source large models to achieve the best performance. The inference speed depends on the network quality. Besides, the API calls produce cost. In the future, we will try to design a data generation pipeline to overcome data scarcity and develop an end-to-end model for generic instruction navigation.

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
