# OpenReview forum: "InstructNav: Zero-shot System for Generic Instruction Navigation in Unexplored Environment"
_robot-learning.org/CoRL/2024/Conference — CoRL 2024_

### Official Review · Reviewer_nMyn · 2024-07-21
**The proposed method requires no additional training and can be applied to common robot platforms. Impact can be higher if considering rebuttal questions.**

**Originality:** 3
**Technical Quality:** 3
**Clarity Of Presentation:** 4
**Potential Impact:** 3
**Recommendation:** 3
**Confidence:** 3

**Review:**

Quality: The paper presents a high-quality research contribution in the field of robotic navigation. The methodologies introduced, namely Dynamic Chain-of-Navigation (DCoN) and Multi-sourced Value Maps, are robust and well-detailed. The experimental results are thorough, demonstrating the system's effectiveness and superiority over existing methods. The use of real-world validation adds significant credibility to the proposed system's practical applicability.

Clarity: The paper is clear and well-organized. Each section logically follows from the previous one, and the figures and tables provided help in understanding the complex concepts. The explanations of DCoN and the Multi-sourced Value Maps are particularly clear, allowing readers to grasp how these components work together to achieve zero-shot navigation.

Originality: The originality of this work is high. The idea of unifying various navigation strategies through a dynamic and adaptable planning paradigm is innovative. The integration of large language models (LLMs) with robotic navigation in a zero-shot context is a novel approach that pushes the boundaries of what current systems can achieve.

Significance: This work is highly significant for the field of robotic navigation and human-robot interaction. By enabling robots to follow diverse natural language instructions without prior training or pre-built maps, this system significantly expands the potential applications of autonomous robots in real-world environments. The improvements in zero-shot performance over task-trained models highlight the practical implications and potential for widespread adoption.

Strengths

Innovative Methodologies: The introduction of DCoN and Multi-sourced Value Maps provides a novel approach to handling diverse navigation tasks in a unified manner.
Zero-shot Performance: The system's ability to outperform trained models in a zero-shot setting is a major strength, demonstrating the robustness and flexibility of the approach.
Real-world Validation: Experiments conducted in real-world settings across various indoor environments add significant weight to the system's practical applicability.
Clarity and Organization: The paper is well-structured and clearly written, making complex concepts accessible to readers.
Comprehensive Evaluation: The thorough experimental evaluation, including ablation studies and comparisons with state-of-the-art methods, provides strong evidence for the system's effectiveness.
Weaknesses

Dependence on Large Models: The reliance on large, possibly closed-source models like GPT-4 could be a limitation, particularly regarding accessibility and deployment costs.
Inference Speed: The system's performance depends on the quality of the network and the speed of inference, which may be a bottleneck in some real-world applications.
Limited Discussion on Failures: While the paper discusses success rates and improvements, there is limited analysis of failure cases and the specific challenges encountered in those instances.
Potential Cost: The API calls for large language models can incur significant costs, which might be a barrier for some applications.

**Quality Of The Limitations Section:**

3

**Questions For Rebuttal:**

Please consider answering the following questions:
Why is it necessary to select several discrete actions instead of continuous linear and angular velocities?
Has the authors consider using panorama views instead of taking images multiple times for a single inference? Would be great to add a quick ablation in the short rebuttal phase.
If there are multiple floors that robot should go to, are there ways to modify current maps to navigate robots to go to different floors?

**Robotics Focus:**

3

**Summary Of Paper:**

This paper introduces a system enabling robots to navigate diverse natural language instructions without prior training or maps. It presents Dynamic Chain-of-Navigation (DCoN), which unifies planning by dynamically updating navigation plans based on observed objects and environment context. Additionally, Multi-sourced Value Maps convert linguistic plans into actionable robot trajectories, incorporating action, semantic, trajectory, and intuition maps for comprehensive guidance. InstructNav achieves notable zero-shot performance on multiple tasks, outperforming several trained models, and demonstrates robustness in diverse real-world settings, such as apartments, offices, and libraries. This system marks a significant advancement in robotic navigation, offering versatile and effective operation across various environments and instructions without specific training.

**Summary Of Recommendation:**

The paper presents a generic instruction navigation method without training any model. This method can be easily applied to multiple platforms, and can have more impact if authors illustrate more about the above rebuttal questions

---

### Official Review · Reviewer_dRHf · 2024-07-21
**Good versatile approach for a variety of language-based navigation tasks**

**Originality:** 3
**Technical Quality:** 3
**Clarity Of Presentation:** 4
**Potential Impact:** 3
**Recommendation:** 3
**Confidence:** 4

**Review:**

Strengths:
- Flexible and novel approach that can handle multiple types of navigation tasks without task-specific training or pre-built maps
- Both the Dynamic Chain-of-Navigation (DCoN) and Multi-sourced Value Maps are shown to have significant benefits in unifying  different navigation instructions and converting linguistic planning into actions
- The authors evaluate InstructNav across multiple different benchmarks and show that they are able to get results that are competitive with SOTA zero-shot methods, even beating them in terms of success in ObjectNav and for almost all metrics in DDN.
- The paper has a generally good structure and I can understand the key components and the rationale/inspiration behind various design choices.

Weaknesses:
- InstructNav relies on text-only and multi-modal large language models like GPT-4 and GPT-4V, which can make it impractical for certain scenarios (due to computational costs or latency/network issues).
- While real-world experiments are conducted, quantitative results from these experiments are not provided; this makes it difficult to assess real-world performance.
- Limited discussion of failure modes: authors do not provide a comprehensive analysis of the system's failure modes or limitations. Understanding when and why the system fails is important for assessing reliability and identifying areas for improvement. Leaving this information out makes it difficult to fully evaluate the robustness of the approach and its potential limitations in real-world scenarios. In particular, for both the ObjectNav and InstructNav tasks, the SPL is significantly lower than other methods with comparable success rates, implying that InstructNav does not take paths that are as efficient. Why is this the case?

Quality: I think the paper is good-quality research with good methodology, comprehensive experiments, and good analysis. The authors have clearly put significant effort into developing a novel approach and evaluating it.

Clarity: The paper is generally well-structured and clearly written.

**Quality Of The Limitations Section:**

3

**Questions For Rebuttal:**

The paper lacks a discussion of failure modes. Please provide a detailed analysis of scenarios where InstructNav struggles or fails, including examples from both simulated and real-world experiments. It is important to show how the design choices made by the authors result in behavior that is distinct from previous methods, both in cases that result better performance than other methods and worse performance.

Clarification needed on ablation study methodology:
Table 4 presents an ablation study of the InstructNav system, including a row 'w/o Dynamic CoN (DCoN)'. However, the paper describes the Dynamic Chain-of-Navigation as necessary for generating inputs for the Multi-sourced Value Maps, particularly the Action Value Map and Semantic Value Map. It's unclear how these maps could be utilized without DCoN.
Please address the following:
- Explain how the system functions without DCoN in this ablation.
- Clarify how inputs for the value maps are generated in the absence of DCoN.
- If there's an alternative method being used, please describe it in detail.
- If there's an error in the table or its description, please correct and discuss any potential impact on the interpretation of results.
For the ObjectNav results, the authors state that their results are “comparable to the best trained object navigation model OVRL”, however I believe that OVRL is a relatively older method that is no longer the ‘best trained model’. From what I remember, PIRLNav outperformed OVRL, and got 70.4% success and 34.1% SPL (vs. 58% and 20.9% from InstructNav).

**Robotics Focus:**

4

**Summary Of Paper:**

InstructNav is a system for generic instruction navigation in unexplored environments, designed to execute diverse language instructions without task-specific training or pre-built maps. Its key innovation, Dynamic Chain-of-Navigation (DCoN), unifies various navigation instructions into a standard planning paradigm, while Multi-sourced Value Maps convert linguistic plans into actionable robot trajectories. The system excels in Object Goal Navigation, Visual Language Navigation, and demand-driven navigation tasks, outperforming existing methods in simulations and real-world experiments. InstructNav demonstrates superior performance on HM3D object goal navigation, achieves zero-shot performance on R2R-CE visual language navigation, and surpasses state-of-the-art on demand-driven navigation. It was tested extensively in simulations and with a Turtlebot 4 in diverse indoor environments.

**Summary Of Recommendation:**

InstructNav is a flexible approach for embodied navigation that handles multiple task types without task-specific training or pre-built maps. The core innovations - Dynamic Chain-of-Navigation and Multi-sourced Value Maps - show potential in unifying navigation instructions and converting linguistic planning into actions, attaining competitive performance with state-of-the-art zero-shot methods across multiple benchmarks. While the paper is well-structured and clearly explains key components, it has limitations including reliance on large language models, lack of quantitative real-world results, and limited discussion of failure modes. To strengthen the work, the authors should provide a detailed analysis of failure modes, and clarify the ablation study methodology (particularly regarding how the system functions without Dynamic Chain-of-Navigation).

---

### Official Review · Reviewer_3AfH · 2024-07-21
**Interesting approach and great results, but I have questions about what kind of commands it can handle**

**Originality:** 4
**Technical Quality:** 4
**Clarity Of Presentation:** 4
**Potential Impact:** 3
**Recommendation:** 3
**Confidence:** 4

**Review:**

Overall this paper is clear and easy-to-read, and provides a novel and interesting approach to handling a wide-range of diverse navigation instructions. The method of using an LLM to decompose the natural language instruction into common navigation commands that can be parameterized by landmarks is nice, and the usage of composed value maps that are, in part generated by a VLM, is particularly nice and a good way to bridge the gaps between LLMs/VLMs and more classical planning-based techniques. The simulation and hardware experiments are also thorough and convincing, and appear to be well-executed.

My biggest concern for this paper is that I think it may not be able to handle a large set of types of commands, such as explicitly avoiding landmarks (“Go to the kitchen but avoid going near the tables”), commands that involve back-tracking (“go to the sofa, then go to the chair, then go back to the sofa”), and cyclic commands (“continuously patrol around the room”). The reason for these concerns is that it is not clear that any of the value maps would be able to effectively capture the first type of command, and that the trajectory-value map would explicitly prevent the second and third kind of behavior. It is also not clear to me how the method would handle instructions that involve spatial relationships in the frame of the landmarks (“Go to the left side of the bed”).

I would recommend accepting this paper conditioned on the authors clarifying how their system would address these kinds of commands.

**Quality Of The Limitations Section:**

2

**Questions For Rebuttal:**

As mentioned in the main review: How does this system handle the following type of instructions:
- Avoiding landmarks (“Go to the kitchen but avoid going near the tables”)
- Instructions involving back-tracking (“go to the sofa, then go to the chair, then go back to the sofa”)
- Cyclic commands (“continuously patrol around the room”)
- Object-centric spatial references  (“Go to the left side of the bed”).

- If a user instructed the robot to move a specific amount forward (e.g: 5 feet forward, 10 meets to the right), how does the action value map take the numeric component into account? Are the locations relative to the robot that are assigned 1 in the move forward / turn right / etc. hard-coded, or are the distances extracted from the instruction?

**Robotics Focus:**

4

**Summary Of Paper:**

This paper proposes InstructNav, an approach to robot navigation that can handle generic instructions in unexplored environments. InstructNav uses Dynamic Chain-of-Navigation to enable a LLM to decompose a generic navigation instruction expressed in natural language into a chain of common navigation actions along with landmarks, and then uses the action and landmarks to generate multiple value maps that are composed into a cost function for a planner to generate navigation paths that accomplish the task.

**Summary Of Recommendation:**

I like this approach and the paper overall, but I do have some concerns about what kind of commands are possible with the current proposed approach.

---

### Author Rebuttal · Authors · 2024-08-12

Here, we upload a ZIP file containing a PDF file and a mp4 Video file.

The PDF file includes supplementary ablation experiments, quantitative real-robot experiments, case studies of simulation and real-robot experiments, baseline comparisons, and so on.

The Video file contains all the successful trajectories from the quantitative real-robot experiments.

---

### Decision · Program_Chairs · 2024-09-04

**Decision:**

Accept

**Comment:**

# Strengths
1. The paper introduces a novel zero-shot method for handling diverse navigation instructions.
1. The real-world demonstrations showcase the effectiveness of the approach.
1. The authors evaluated InstructNav across multiple benchmarks.
1. The paper provides a comprehensive evaluation, including ablation studies.

# Weaknesses
1. Although real-world experiments are conducted, quantitative results from these experiments are missing.
1. The proposed method relies mainly on multimodal large language models, which can be impractical due to computational costs, latency, and network issues.
1. Failure mode analysis is insufficient.

### Post-rebuttal comment
The reviewers agree in their recommendation to accept the paper because most of the concerns have been addressed. I agree with their consensus.